# Early Detection of Corrosion-Induced Concrete Micro-cracking by Using Nonlinear Ultrasonic Techniques: Possible Influence of Mass Transport Processes

**Miguel-Ángel Climent** [1,*], **Marina Miró** [1], **Jesús-Nuño Eiras** [2], **Pedro Poveda** [3], **Guillem de Vera** [1], **Enrique-Gonzalo Segovia** [1] **and Jaime Ramis** [3]

1    Civil Engineering Department, University of Alicante, Sant Vicent del Raspeig, 03690 Alicante, Spain; m.miro@ua.es (M.M.); guillem.vera@ua.es (G.d.V.); enrique.gonzalo@ua.es (E.-G.S.)
2    Aix Marseille Univ., CNRS, Centrale Marseille, LMA, Marseille, France; jesus.eiras.fernandez@gmail.com
3    DFISTS, University of Alicante, Sant Vicent del Raspeig, 03690 Alicante, Spain; pedro.poveda@ua.es (P.P.); jramis@ua.es (J.R.)
*    Correspondence: ma.climent@ua.es

**Abstract:** This work presents results allowing an unequivocal correlation of the observations of strong nonlinear elastic features of ultrasonic waves (values of the nonlinearity parameters exceeding the thresholds corresponding to undamaged states), with the critical events of the corrosion-induced surface cracking of reinforced cement-mortar specimens. These observations point to the possibility of the early detection of cracking using nonlinear ultrasonic (NLU) techniques. Experimental evidence is presented on the existence of active net mass transport processes, due to wick action, in the course of the corrosion tests, in the experimental conditions of this work. These phenomena might explain the observed abrupt shifting of the nonlinear parameter values (typically increasing and then decreasing post-peak, even reaching values typical of the undamaged state), and, partially, the high variability obtained for the values of the nonlinear parameters in damaged (cracked) states. Finally, some consequences are derived from the point of view of use of the NLU techniques in engineering practice, i.e., in surveys aimed at evaluating reinforced concrete structures affected by corrosion.

**Keywords:** reinforced concrete; cement mortar; steel corrosion; damage; cracking; nonlinear ultrasonic test; non-destructive detection

## 1. Introduction

The corrosion of embedded steel is one of the main threats to the durability of reinforced or pre-stressed concrete structures [1]. The damage to concrete, resulting from the corrosion of steel reinforcement, is a consequence of the expansive character of oxide formation [2]. The damage manifests in the form of micro- and macro-cracking [3,4], and eventual localized or generalized spalling of the concrete cover. In addition to the loss of cover, a reinforced concrete member can suffer structural damage due to the loss of bond between the steel and concrete, and loss of rebar cross-sectional area [5]. Many authors [6], and most concrete model codes [7], usually consider the appearance of cracking as the limit state regarding the durability of concrete structures suffering from reinforcement corrosion.

The survey and inspection of structures is an important issue related to the field of maintenance and repair of constructions affected by reinforcement corrosion. In this sense, non-destructive appraisal techniques able to provide an early warning for corrosion initiation, or corrosion risk, are of much interest. There is a wide experience of using several classical electrochemical techniques for assessing the state and risk of the corrosion of steel embedded in concrete [8–11]. The capability of the non-destructive electrochemical techniques to identify the development of corrosion at a very early stage, before the concrete cover has suffered any physical damage, is remarkable. Nevertheless, these techniques are not sensitive to the micro-cracking induced in concrete by the steel corrosion process.

Elastic wave methods have been used for assessing mechanical properties and for detecting damage in materials and structures. Waves propagating through a material, such as concrete, are reactive to defects or discontinuities, such as cracks, voids, or even corrosion products [12]. The major wave-based methods for corrosion damage assessment in reinforced concrete structures are: ultrasonic pulse velocity (UPV), impact echo (IE), and acoustic emission (AE). The UPV technique, which involves the generation of guided modes in the steel reinforcement bar, is sensitive to the nature of the interface between steel and concrete, and also to cracking [13]. Nevertheless, the UPV measurements through concrete, which would be much more useful from the point of view of non-destructive testing, show low sensitivity to distributed micro-cracks, which appear during the initial stages of steel corrosion in concrete [14]. The IE technique, which relies on using a mechanical impact and then registering the signal received by sensors placed on the concrete surface, has been scarcely used for the detection of damage due to corrosion [15]. The reliability of the IE method decreases with an increase in the thickness of the concrete members [12]. The AE technique has been widely used for detecting active deformation processes, such as crack growth, void closure, plastic deformation, corrosion, and other degradation phenomena [12,16]. Written recommendations for the application of the AE method to crack detection and damage evaluation in concrete even exist [17]. However, an important limitation of the AE method is that it cannot effectively detect passive defects [12], thus reducing its applicability for occasional surveys of existing reinforced concrete structures.

During the last decades, nonlinear ultrasonic (NLU) techniques have been shown to be useful as new non-destructive methodologies for the evaluation of material degradation [18,19]. In concrete, the interaction of the elastic waves with defects and heterogeneities generates several nonlinear acoustic phenomena, which can be exploited for non-destructive damage assessments [20]. Among these phenomena we can find amplitude-dependent resonant frequency and attenuation shifts [21–23], higher harmonic generation [24–27], and amplitude and frequency wave modulations [28–30].

The NLU techniques have been used to investigate damage progress during the loading tests of granite samples [31], concrete specimens [32,33], and experimental models of real field reinforced concrete members [34]. Several publications have reported the application of these techniques to study thermal damage in sandstone [35] and concrete [22,36]. The drying shrinkage of concrete was also studied, with acoustic nonlinear parameter finding an increasing trend of nonlinearity due to the formation of tensile stress on the dry surface [37]. The technique of Nonlinear Impact Resonance Acoustic Spectroscopy (NIRAS) has been shown to allow the characterization of the damage state of cement-mortar samples affected by an alkali–silica reaction [38], and it has even shown promise as a possible method of rapid identification of alkali-reactive aggregates [39]. NIRAS is also suitable for non-destructively monitoring the aging of alkali-resistant glass-fiber-reinforced cement [40], and for detecting damage at early ages during external sulphate attack tests on Portland-cement-mortar samples [41].

Only a few works can be found reporting the application of NLU techniques in reinforced concrete corrosion tests [42–48]. It was shown that both the higher harmonic generation [43,44], and the frequency modulation [42,44] techniques, were able to successfully differentiate between undamaged and damaged states of reinforced concrete specimens after a number of cycles of accelerated corrosion of the embedded steel coupled with wetting–drying cycles. Similarly, Antonaci et al. showed the higher sensitivity of nonlinear ultrasonic measurements performed using the Scaling Subtracting Method [45], as compared to the linear indicators (based on velocity or attenuation measurements), for identifying small variations of the microstructure due to the cracking of a concrete specimen after sequential cycles of wetting and accelerated corrosion of the reinforcing steel. Some recent publications [46–48] have reported results of applications of nonlinear ultrasonic techniques (higher harmonic generation and frequency modulation) to the continuous study of the micro-cracking process since the beginning of the electrically accelerated corrosion tests. This requires a parallel monitoring of the nonlinear ultrasonic indicators and

some physical property directly related to the progress of cracking, such as the observation of the appearance of the first visible micro-crack due to corrosion, and the evolution of its crack width [3,49–51]. Another requirement of this type of study [46–48] is maintaining a practically stable and adequate moisture state of concrete in order to avoid the need to apply wetting–drying cycles of the reinforced concrete specimens. It is known that the water saturation level of the reinforced concrete specimen might have a strong influence on the nonlinear ultrasonic indicators, especially at high levels of corrosion [45,52].

It was shown that the appearance of surface micro-cracks due to steel corrosion in reinforced cement-mortar specimens seems to be preceded and accompanied by the observation of strong nonlinear features in the received signals [46–48]. However, after these critical events of the cracking process the nonlinear indicators tend to return to values typical of the pre-critical situation [46,47]. Another observation is that the corrosion-induced damage statistically increased both the range of measured values and the variability of a relevant nonlinearity parameter, such as the Intensity Modulation Ratio (*R*) in Nonlinear Wave Modulation Spectroscopy measurements [48]. A hypothesis was advanced regarding the possible effect of the filling of the void space with liquid containing steel corrosion products after the formation of new cracks or enlargement of its width [47,48]. It is reasonable to admit that any process leading to filling or closing the pre-existing cracks would affect the interaction of the elastic wave with the defective medium [45,53], possibly reducing the capability of the NLU techniques to effectively detect the cracks, and also contributing to the observed variability of the nonlinearity parameters [48]. The aim of this work is to present further evidence of the correlation of the observed increases in the nonlinearity parameters with the relevant events of the cracking of cement mortar due to corrosion of the steel reinforcement. Additionally, other observations will be presented regarding the existence of dynamic convective mass-transport processes of liquid (dragging steel corrosion products) during the accelerated corrosion tests performed in the experimental conditions of the previous publications [46–48]. Finally, we discuss the implications of the abovementioned findings (shiftings and variability of values of the nonlinearity parameters) on the possibility of applying the NLU techniques to the practical detection of cracks in reinforced concrete structures affected by corrosion.

## 2. Materials and Methods

This work contains the results of two experimental campaigns of electrically accelerated corrosion tests of steel-reinforced cement-mortar (RCM) specimens, during which NLU measurements were obtained. The first set of experiments, performed on three RCM specimens (named here as samples 1, 2, and 3), was aimed at finding the correlations between the appearance of strong nonlinear features in the received wave signal and the critical events of the corrosion-induced cracking process, especially during the stages of generation and coalescence of micro-cracks, and the early development of the width of the first visible surface cracks [3]. A detailed description of the experimental setup and procedure corresponding to this first set of experiments can be found in a previous publication [47]. The second set of experiments, performed on five RCM specimens (named as samples M0, M1, M2, M3, and M4), was aimed at determining if the IMD technique was able to clearly identify different levels of accrued damage due to corrosion. In this way different durations of the electrically accelerated corrosion were applied, and the NLU survey was extended with measurements obtained some time after ceasing the passing current and allowing some drying of the RCM specimens. A detailed description of the experimental setup and procedure corresponding to this second set of experiments can be found in a previous publication [48].

### 2.1. Materials and Sample Preparation

All RCM specimens were prismatic with dimensions of $100 \times 100 \times 350$ mm$^3$, and were prepared with the same composition detailed in Table 1. Standard siliceous sand and a sulphate-resisting Portland cement with a high speed of mechanical strength gain, CEM I

52,5 R SR(3) [54], were the main components of the mix. The water-to-cement ratio (w/c) was set at 0.5. An amount of NaCl dissolved into the mixing water allowed us to obtain a content of 2% Cl$^-$ relative to cement mass in the hardened mortar [55,56], thus practically ensuring 100% current efficiency in the experimental conditions of the accelerated corrosion test [57]. The fresh cement mortar was poured into plastic molds of the adequate dimension, mechanically compacted (manual compaction), and cured in a humidity chamber at 20 °C and 95% relative humidity. The curing times were of 7 days for the first set of experiments (specimens 1, 2, and 3), and of 28 days for the second set of experiments (specimens M0, M1, M2, M3, and M4). Once the curing periods were finished, the accelerated corrosion tests and the NLU measurements were started.

**Table 1.** Composition of the cement mortar.

| Material | Amount (g) |
|---|---|
| Cement (CEM I 52,5 R SR(3)) | 450 |
| Standard siliceous sand | 1350 |
| Deionized water | 225 (w/c = 0.5) |
| NaCl | 14.8 (2% Cl$^-$ relative to cement weight) |

The molds allowed for the center crossing of a smooth steel rebar of a 12 mm diameter along each sample. The cement-mortar cover depths over the steel bar were 10 mm and 25 mm for the first and second sets of experiments, respectively (see Figure 1). This layout was chosen to promote the appearance of the first visible crack on top of the upper surface of the RCM specimens, thus allowing for the monitoring of the evolution of crack width with time. The steel bars were previously cleaned from native corrosion products following a recommended procedure [58] and weighted, covering the ends with vinyl electric tape to avoid the steel–mortar–air contact.

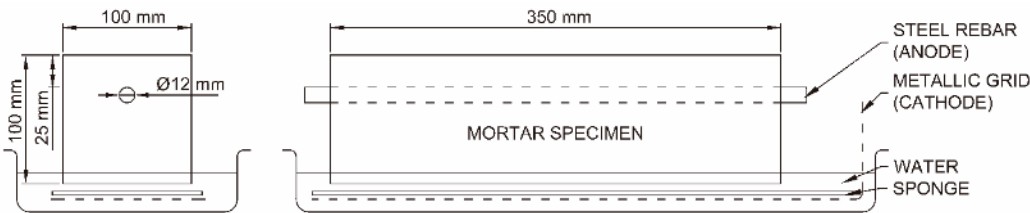

**Figure 1.** Mortar-specimen dimensions corresponding to the second set of experiments (specimens M0, M1, M2, M3, and M4) and the anode/cathode disposition. In the case of the first set of experiments (specimens 1, 2, and 3), the mortar cover depth over steel was 10 mm instead of 25 mm. Figure adapted from [48].

### 2.2. Accelerated Corrosion Tests

The accelerated corrosion tests were conducted under galvanostatic conditions, using a potentiostat-galvanostat (Model 362, EG&G Instruments, Princeton, NJ, USA). The steel rebar was set as the anode and an external galvanized steel grid was used as the cathode (see Figure 1). To keep an adequate level of electric conductivity, the samples were partially submerged (5 mm height) in a tray filled with tap water, and a polypropylene sponge was placed between the mortar specimen and the cathode, Figure 1. The constant anodic current densities were 40 and 100 µA/cm$^2$ for the first and second sets of experiments, respectively.

The physical damage of the RCM specimens due to corrosion was followed by detecting the appearance of the first surface micro-crack, and by monitoring the evolution of the crack width over time (daily measurements). The crack-width measurements were performed using a microscope (magnification 40×, model 58-C0218, Controls, Milan, Italy), with a limit of detection of 10 µm. Further details of the experimental setup and procedures of the accelerated corrosion tests can be found in the previous publications [47,48].

### 2.3. The Assessment of Material Damage through Nonlinear Elastic Wave Features

The degradation of a material, for instance, due to cracking, is linked to a nonlinear mechanical behavior [59]. When an ultrasonic wave propagates through the material and interacts with the microstructural defects, nonlinear terms should appear as linked to these elastic waves. As the micro-damage increases, nonlinearity should increase. In this work, the NLU measurements were obtained through techniques based on higher harmonic generation, here termed as harmonic distortion (HD), and frequency wave modulation, here termed as intermodulation distortion (IMD) [60] (see Figure 2). HD takes place when a nonlinear system is excited by a signal of frequency $f_1$. In this case, higher-order frequencies appear at the output ($f_2 = 2f_1, f_3 = 3f_1$, and so on), Figure 2a. Regarding IMD, if a nonlinear system is excited by two signals of frequencies $f_0$ and $f_1$, referred to as pump ($f_0$) and probe ($f_1$) waves, being $f_1 \gg f_0$, the mechanical nonlinearity produces additional output-frequency components as sum and difference as $f_1 \pm n \cdot f_0$ (for n = 1, 2, ... , N) (see Figure 2b). Otherwise, if the system behaves linearly, the sideband frequencies are not generated.

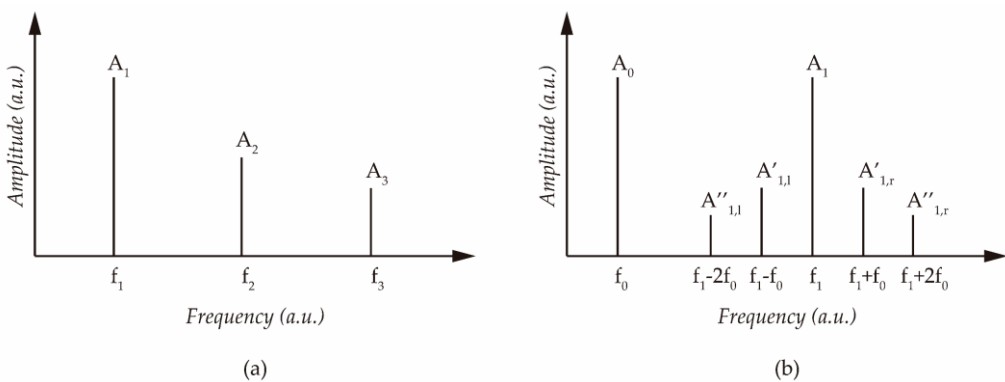

**Figure 2.** Frequency spectrum: (**a**) harmonic distortion (HD) and (**b**) intermodulation distortion (IMD). Figure adapted from [47].

Quantitative nonlinearity parameters can be derived from the amplitudes of the fundamental and harmonic frequencies in the case of HD experiments. Assuming that changes in the wave propagation velocity and the attenuation are small, and in experimental conditions similar to those used in this work (NLU measurements performed with an input signal of fixed frequency, and the transducers always located at the same positions), the parameters of nonlinearity can be approximated [24,61,62] as the harmonic ratios $\beta = A_2/A_1^2$ and $\beta' = A_3/A_1^3$, where $A_1$, $A_2$, and $A_3$ are the amplitudes of the fundamental, second-, and third-harmonic waves, respectively (see Figure 2a). Taking into account that, usually, the odd-numbered harmonics have higher amplitudes than the even-numbered ones, and that the third harmonic ratio is much more sensible to the micro-damage caused by corrosion [43,44,47], in this work, only the results corresponding to the third harmonic ratio ($\beta'$) are presented. In practice, the variations of these ratios relative to the initial undamaged condition are used as the indicators of the microstructural damage [24].

As for the specific parameters that have been used to quantify the damage by incorporating intermodulation products, the intensity modulation ratio, $R$, is defined as [62]:

$$R = \frac{\sum_{n=1}^{N} \left( A_{left}^n + A_{right}^n \right)}{A_1} \tag{1}$$

where $A_{left}^n$ and $A_{right}^n$ are the amplitudes of the $N$ sideband intermodulation products, and $A_1$ is the amplitude of the fundamental high-frequency probe. In many studies, $R$ has been used as the damage index [63–65].

Recently, it has been proposed using a new parameter for putting in evidence the micro-damage through the intermodulation nonlinear features. This new parameter, termed

as *DIFA* (difference of amplitudes) (Equation (2)), is considered to represent the redistribution of elastic energy between the fundamental frequency component and the various intermodulation products generated when the elastic wave travels through the medium in the presence of critical micro-defects [47].

$$DIFA = A_1 - \sum_{n=1}^{N} \left( A_{left}^n + A_{right}^n \right) \qquad (2)$$

In this work, only the first-order ($A_{f1\pm f0}$) and second-order ($A_{f1\pm 2f0}$) intermodulation products were considered for calculating the values of both the *R* and the *DIFA* parameters, i.e., *N* is set to 2 (see Figure 2b).

*2.4. Nonlinear Ultrasonic Measurements*

In the first series of experiments (specimens 1, 2, and 3), both HD and IMD measurements were performed with a "direct-transmission" mode, i.e., with ultrasonic transducers located at opposite faces of the RCM specimen. The positions of the transducers were equidistant to the rebar axis and to the specimen's lateral surface (same height from the specimen's bottom) (see Figure 3). Both transducers were permanently glued (at the beginning of the NLU measurement series) to the RCM specimen, using a quick-setting Cyanoacrylate glue as the coupling agent. The tests were conducted with an NI-USB 6361 multifunction I/O device with a sampling frequency of 2 MHz and an ACD resolution of 16 bits. For the HD experiments, a 30 kHz sinusoidal signal with a length of 10,000 cycles was sent to an FS WMA-100 voltage amplifier and then to the emitter transducer. Amplitudes between 120 to 200 V in 10 V steps were used for the excitation signal. The received signal was amplified using a signal conditioner 2693-A (Brüel & Kjaer, Naerum, Denmark) and then sent to the acquisition platform. For the IMD experiments, the emitter transducer was supplied with two tones at the same time: $f_1$ = 30 kHz (probe wave) and $f_0$ = 2 kHz (pump wave). IDK09 transducers [66] with an isolated contact membrane ($Al_2O_3$ pure ceramic) and piezoelectric ceramic active element $PbZrTiO_3$ (PZT)—modified ceramic class 200—were used for both the emitter and receiver elements. A rectangular window was applied to the steady-state interval of the received signal. The frequency spectrum of the windowed signal was obtained using the Fast Fourier Transform method. Then, the amplitudes of the fundamental (corresponding to 30 kHz), second- (60 kHz), and third-harmonic (90 kHz) waves were determined. A similar process was performed for the intermodulation products (first-order-intermodulation products are expected at 28 and 32 kHz and second-order products near 34 and 26 kHz). Further details of the experimental procedure can be found in the previous publications [47,67].

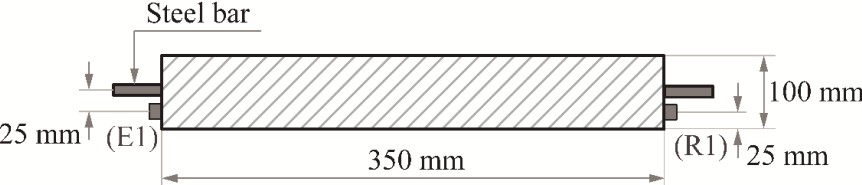

**Figure 3.** Positions of the transducers in the first series of experiments (top view of the RCM specimen). E1: emitter and R1: receiver. Figure adapted from [47].

In the second series of experiments (specimens M0, M1, M2, M3, and M4), IMD measurements were performed with an "indirect-transmission" mode, i.e., with transducers located on a single surface (see Figure 4). It must be considered that this "indirect-transmission" mode might represent the only possibility (only a single surface accessible) when trying to apply the NLU techniques to existing structures or constructive elements, for instance, in the case of reinforced concrete slabs [67]. Figure 4 shows the experimental setup used for the ultrasonic measurements in this second series of tests, and the rela-

tive positions of the emitting (EL and EZ) and receiving (RZ) transducers. The test was repeated at the three different positions shown in Figure 4. Two transducers were used to simultaneously supply two pure tones: $f_0$ = 20 kHz (pump wave) and $f_1$ = 200 kHz (probe wave). The high-frequency probe signal was emitted with a signal generator (SONY AFG310) at an amplitude of 5 V. The same 16 bit ADC resolution I/O device described in the preceding paragraph was used for the generation of the low-frequency pump and the acquisition of the frequency-modulated signal with a sampling frequency of 2 MHz. The acquisition length was set to 50 ms. The pump wave signal was fed through an amplifier FS WMA-100 and then transmitted through a specially designed and manufactured Langevin transducer [48]. These devices have a very narrow bandwidth, making them suitable for providing the needed frequency and energy of the pump wave [48]. The input voltage was set to 130 V (after amplification). Two broadband ultrasonic transducers IDK09 [66] were used for emitting and receiving the high-frequency signal. White soft paraffin (Acofarma) was used as a coupling agent. Further details of the experimental procedure can be found in the previous publications [48,67].

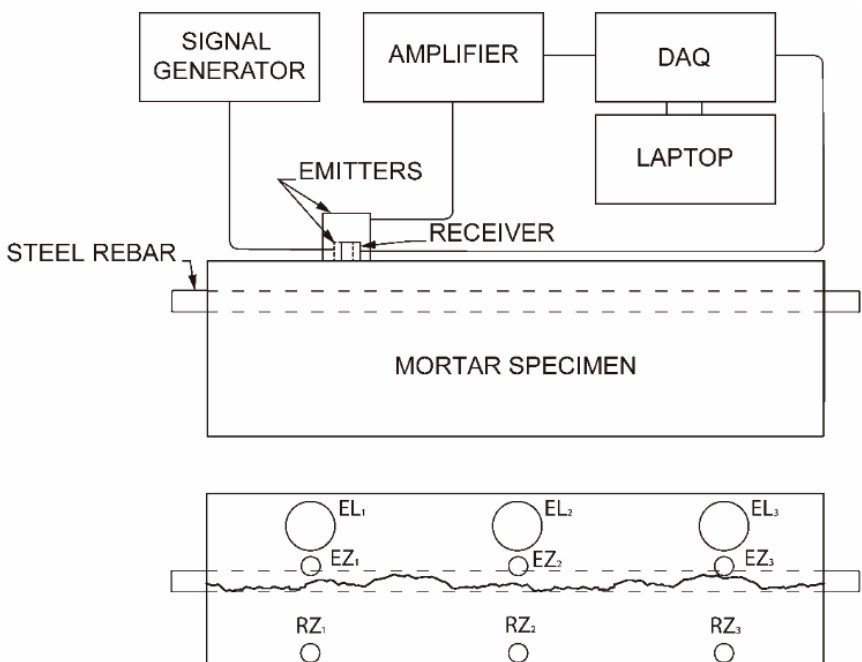

**Figure 4.** Experimental setup used in the second series of experiments, and relative positions of the transducers: Langevin emitter (EL), IDK emitter (EZ), and IDK receiver (RZ). Figure adapted from [48].

## 3. Results and Discussion

The experimental conditions of this work were chosen to allow parallel monitoring of the relevant events of the corrosion-induced damage progress, such as the observation of the first visible surface micro-crack, and the evolution of its crack width, together with the measurement of the nonlinear ultrasonic indicators. Obviously, surface observations and measurements do not provide the full picture of the oxide formation and accumulation at the steel bar and the cracking process of the cement-mortar cover over steel. Nevertheless, some critical events can be identified through careful surface observation. It is reasonable to admit that the first appearance of an open surface crack is preceded by the active development of the incipient micro-cracks through the cementitious matrix, probably starting at the steel–cement paste interface, and coalescing to form a continuous crack, which finally opens to the concrete surface [45,47,67]. In a similar manner, the observation of a sudden considerable increase in the surface crack width also indicates an immediately precedent active development of the inner micro-cracking process. All these critical events

of the cracking process should enhance the nonlinear features of the received ultrasonic waves. At the same time, the other objective of this work was to obtain NLU measurements with different damage states caused by the corrosion of the steel bar in the RCM specimens. Hence, different durations of the current passing period were implemented in these tests.

In the experimental conditions of this work, it was possible to calculate the approximate times of cracking through an empirical expression relating the steel corrosion penetration needed to generate a corrosion-induced surface crack of a width of approximately 50 µm and the ratio between the concrete cover depth over steel (*c*) and the steel bar diameter (∅) [3]. Following this, it was possible to derive an expression for calculating the time needed for generating the abovementioned first surface crack, here termed as $t_{50}$ [67,68]:

$$t_{50} = \frac{236.05 + 292.16\frac{c}{\varnothing}}{I_{corr}} \tag{3}$$

where in $t_{50}$ is expressed in days, *c* and *ø* are expressed in mm, and the corrosion rate ($I_{corr}$) is expressed in µA/cm$^2$.

Table 2 presents the most relevant data of the accelerated corrosion tests corresponding to both the first and the second series of experiments. The sixth column presents the calculated times for generating a surface crack of 50 µm following Equation (3). The seventh column shows the times of passing current, which were chosen in order to obtain different states of corrosion-induced damage, including a control specimen with no artificially induced corrosion (specimen M0), and situations in which the passing current was interrupted before the observation of a surface micro-crack (specimens M1 and M2). The eighth column presents the times at which the appearance of the first surface micro-crack were observed. It is appreciated that the cracking times (seventh column) were slightly lower than the predicted $t_{50}$ values (sixth column). In this regard, it must be considered that the experimental setup used in this work allowed for the observation of surface micro-cracks with a limit of detection of 10 µm (see Section 2.2). Furthermore, the empirical expressions on which Equation (3) is based, were obtained from the experiments performed on reinforced concrete specimens (not cement-mortar specimens), and with a range of *c/ø* values between 2 and 3. In the first series of experiments, the RCM specimens showed the first surface crack between the 7th and 9th days of passing the current, while, in the second series of experiments, only the M3 and M4 specimens presented the first crack on the 6th day, while specimens M0 (control), M1, and M2 did not show any surface cracks. These situations, together with the different durations of the passing-current periods, provided different states of corrosion-induced damage [48]. In all the experiments, the NLU survey was started one day before the onset of the accelerated corrosion test, i.e., at *t* = −1 day. Therefore, the origin of time (*t* = 0) was obtained at the beginning of the passing-current period. The NLU measurement results obtained at *t* ≤ 0 served for providing the reference values of the nonlinear parameters corresponding to an undamaged state [46–48].

**Table 2.** Relevant parameters of the accelerated corrosion tests.

| Series of Experiments | Specimen | *C* (mm) | (c/ø) | $I_{corr}$ (µA/cm$^2$) | $t_{50}$ (Days) | Time of Passing Current (Days) | Time of Cracking (Days) | Duration of the NLU Survey (Days) |
|---|---|---|---|---|---|---|---|---|
| 1 | 1 | 10 | 0.83 | 40 | 12 | 7 | 7 | 8 |
|  | 2 | 10 | 0.83 | 40 | 12 | 14 | 8 | 15 |
|  | 3 | 10 | 0.83 | 40 | 12 | 30 | 9 | 31 |
| 2 | M0 | 25 | 2.1 | 100 | 8.4 | 0 | - | 30 |
|  | M1 | 25 | 2.1 | 100 | 8.4 | 3 | - | 30 |
|  | M2 | 25 | 2.1 | 100 | 8.4 | 6 | - | 30 |
|  | M3 | 25 | 2.1 | 100 | 8.4 | 6 | 6 | 30 |
|  | M4 | 25 | 2.1 | 100 | 8.4 | 20 | 6 | 30 |

### 3.1. Correlation between the NLU Observations and the Corrosion-Induced Cracking Process

In this work, a new form of expression of the corrosion-induced damage, based on the surface observations, was proposed. When daily observations and measurements of the surface crack width were available, it was possible to calculate the derivative of the crack width with time ($dw/dt$), expressed as:

$$\left( \frac{dw}{dt} \right)_i = \frac{w_i - w_{i-1}}{t_i - t_{i-1}} \tag{4}$$

where $w_i$ is the crack-width value measured at the day $i$, and $t_i$ is the time that elapsed since the beginning of the passing-current period. In this way, it is easier to appreciate the most relevant events of the cracking process, i.e., the appearance of the first surface crack and the sudden enlargements of its width.

Figures 5–7 show the correlations between the evolutions of $dw/dt$ and the nonlinear parameters for the reinforced mortar specimens 1, 2, and 3: the harmonic ratio $\beta'$ in Figure 5, the intensity modulation ratio $R$ in Figure 6, and the *DIFA* parameter in Figure 7 (see Section 2.3). The dotted horizontal line in Figure 5 represents the arbitrary threshold considered as indicative of a critical increase in the nonlinear elastic features, i.e., a tenfold increase in the initial value of $\beta'$ recorded at the undamaged state [46,47].

It is apparent from Figure 5 that the harmonic ratio $\beta'$ starts to reach values over the threshold considered as indicative of critical nonlinear features, a few days before the first appearance of a surface crack (Figure 5a,b). In these cases, the first crack observed had an initial width of 20 μm. The situation is slightly different in Figure 5c, where the initial crack width measured is about 10 μm, and its appearance is not preceded by an increase in $\beta'$ over the threshold. Instead, the values of the harmonic ratio started to exceed the threshold about four days before the sudden increase in the crack width recorded on the 20th day ($dw/dt = 40$ μm/day). These observations clearly point to an excellent correlation between the increases in the harmonic ratio and the relevant events of the corrosion-induced cracking. Furthermore, the results also indicate the capability of the NLU technique for providing an early detection of the corrosion-induced damage of mortar and concrete: the non-elastic features can be detected well before the observation of any open surface crack.

The observations of Figures 6 and 7 are also in line with that of Figure 5. The intensity modulation ratio $R$ also clearly starts showing increased values before the first appearance of a surface crack (Figure 6b,c) and before the sudden increases in the crack width observed, for instance, on the 20th day in Figure 6c. The only exception is represented by the behavior observed in Figure 6a. However, it needs to be taken into consideration that the experiment corresponding to Figure 6a was interrupted on the 7th day, just after the observation of the first surface crack and the specimen was broken to observe the incipient crack [47]. Hence, we do not have enough data allowing us to clearly establish the evolution of $R$ in the case of specimen 1. A parallel discussion can be developed on the basis of data in Figure 7a–c. The *DIFA* parameter, defined in Equation (2), starts to decrease its recordings and reaches values close to zero a few days before the first appearance of a surface crack or before the sudden enlargements of the recorded crack width.

Regarding the tests corresponding to specimens M0, M1, M2, M3, and M4 (second series of experiments), Figure 8 shows that their results are also in line with the observations of Figures 5–7. It must be taken into account that the data of Figure 8 correspond only to the central part of the tested specimens (see Figure 4). This means that the NLU data have been obtained with the transducers located in the middle part of the specimen (position 2 in Figure 4), and the crack width measurements have been performed just in this central zone. The rationale behind this decision is to avoid the unavoidable dispersion of results expected, if we mix in the same Figure the results corresponding to the three measuring areas in Figure 4. This dispersion would most probably hide any existing correlation between the evolutions of $dw/dt$ and $R$. The dotted horizontal lines in Figure 8 correspond to a statistically established upper control limit (UCL) for the initial (undamaged) $R$ values [48].

It is appreciable from Figure 8 that the values of *R* hardly obtained values over the UCL in the cases of the specimens that did not show any surface cracks during the tests (specimens M0, M1, and M2). On the other hand, for specimens M3 and M4, *R* values clearly exceeded the UCL a few days before the appearance of the first surface crack, or before any sudden enlargement of the crack width in the central part of the specimens.

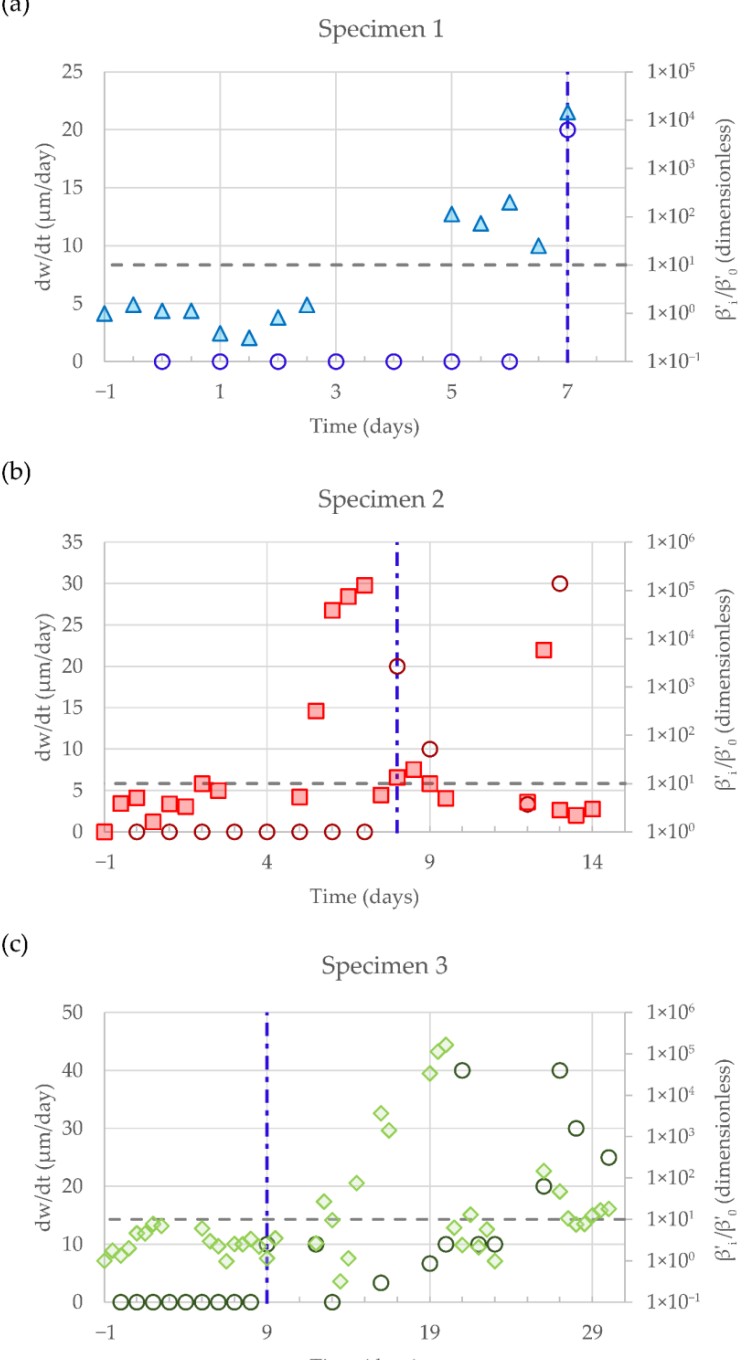

**Figure 5.** Correlation between the evolutions with time of $dw/dt$ (circle symbols) and those of the third harmonic ratio $\beta'$: $\triangle$, $\square$, $\lozenge$. (**a**) Specimen 1; (**b**) specimen 2; and (**c**) specimen 3. The vertical line indicates the time of observation of the first surface crack. The dotted horizontal line represents the arbitrary threshold considered as indicative of a critical increase in the nonlinear elastic features (see text for details).

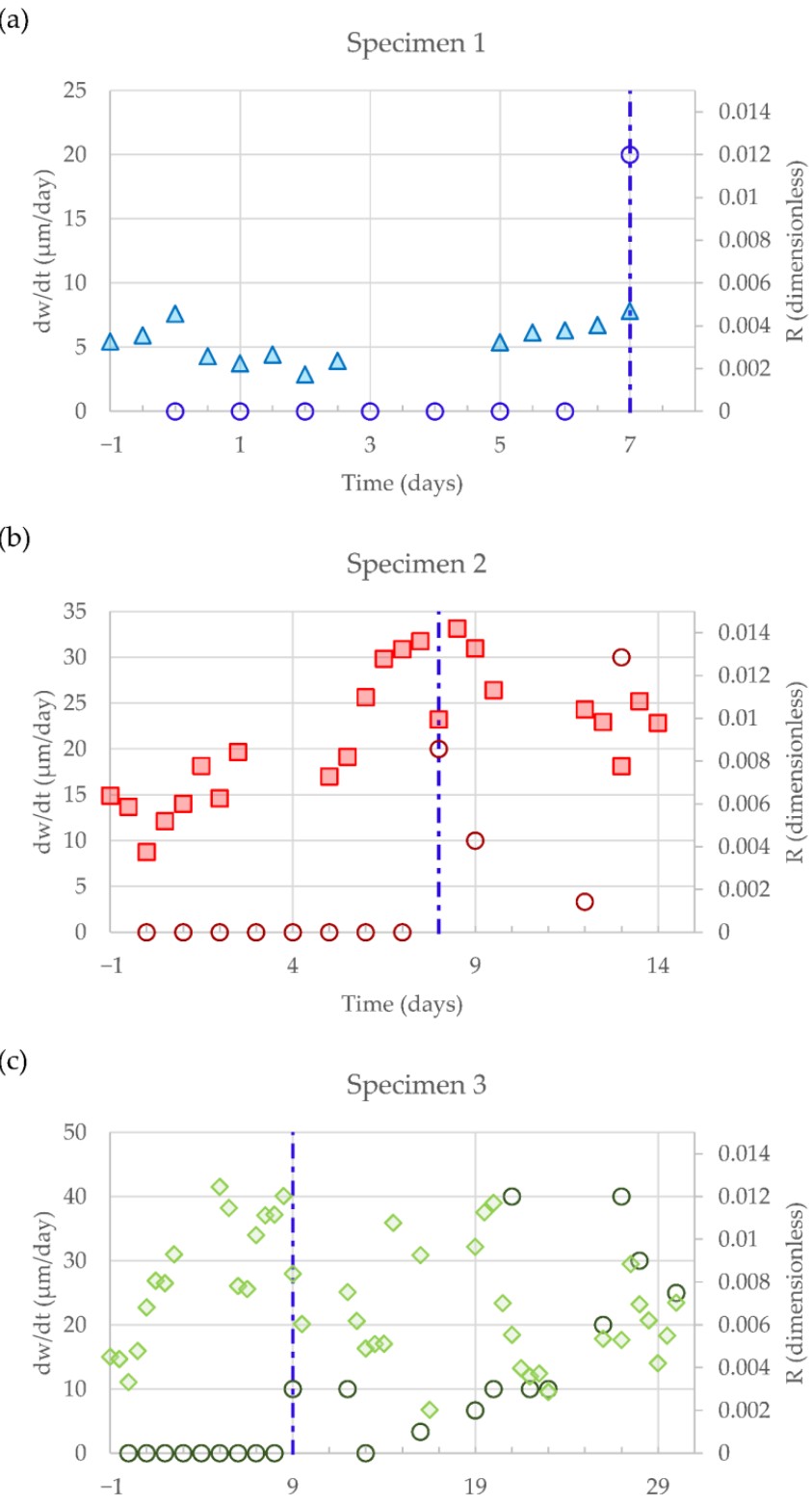

**Figure 6.** Correlation between the evolutions with time of $dw/dt$ (circle symbols) and those of the intensity modulation ratio $R$: Δ, □, ◊. (**a**) Specimen 1; (**b**) specimen 2; and (**c**) specimen 3. The vertical line indicates the time of observation of the first surface crack.

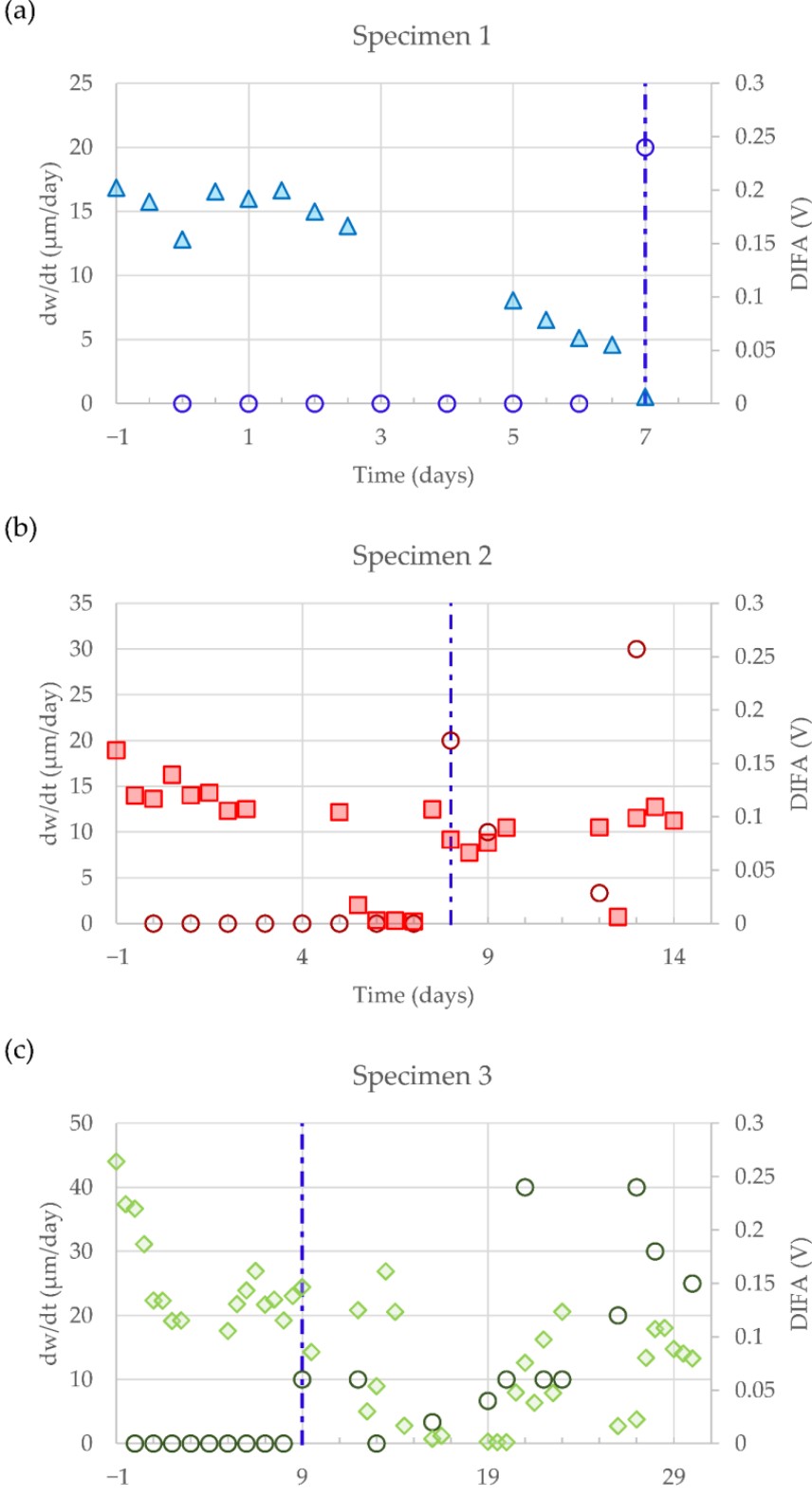

**Figure 7.** Correlation between the evolutions with time of *dw/dt* (circle symbols) and those of the *DIFA* parameter: Δ, □, ◇. (**a** Specimen 1; (**b**) specimen 2; and (**c**) specimen 3. See Section 2.3. The vertical line indicates the time of observation of the first surface crack.

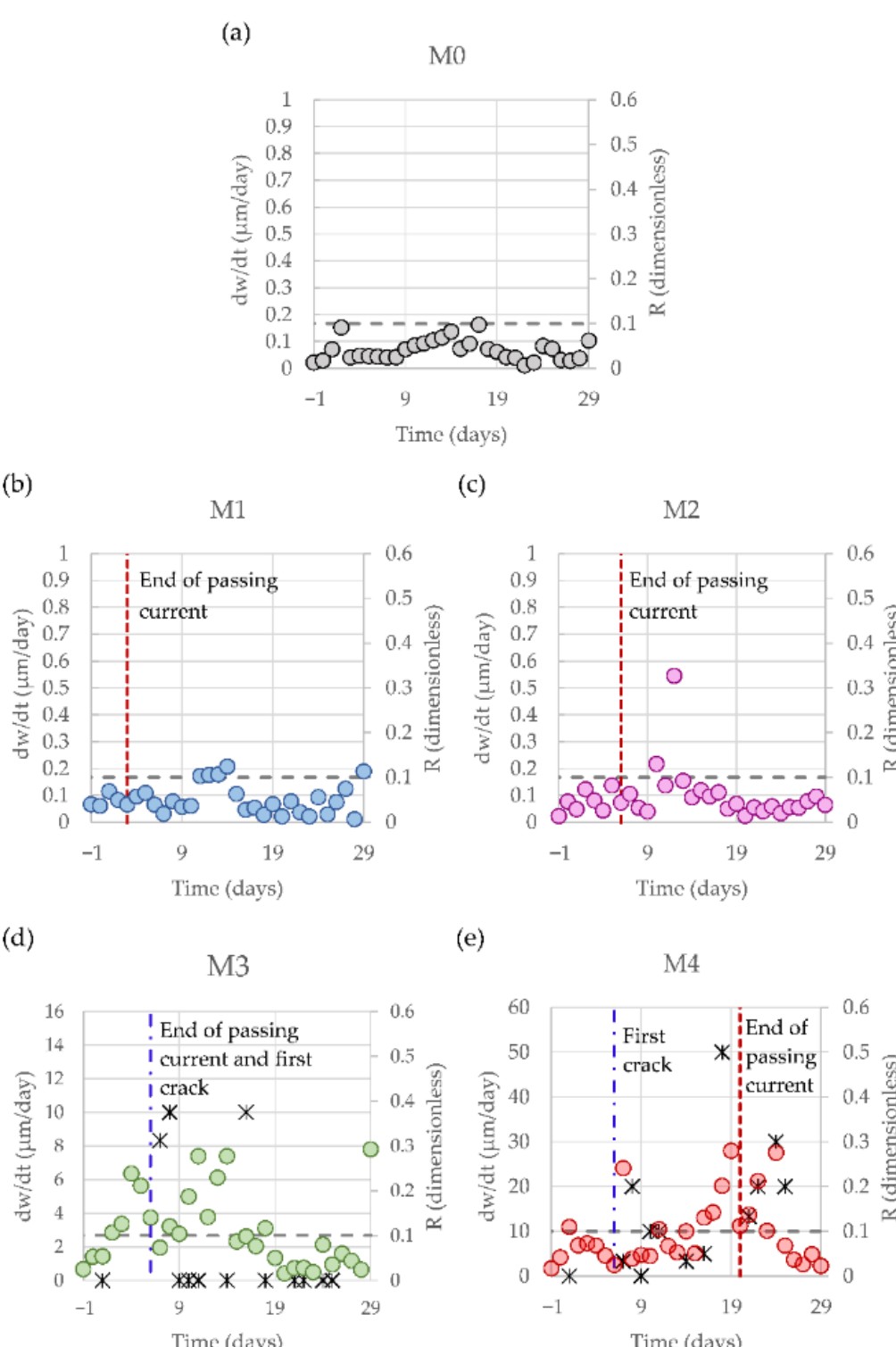

**Figure 8.** Correlation between the evolutions with time of *dw/dt* (* symbols) and those of the intensity modulation ratio *R* (circle symbols). (**a**) Specimen M0; (**b**) specimen M1; (**c**) specimen M2; (**d**) specimen M3; and (**e**) specimen M4. The vertical lines indicate the time of the end of passing current, and the time of observation of the first surface crack, which coincide in the case of specimen M3. The dotted horizontal lines correspond to a statistically established upper control limit for the initial (undamaged) *R* values [48].

The results presented in this subsection clearly point to establishing an unequivocal correlation between the relevant nonlinear features, put in evidence by the nonlinear

parameters of the NLU techniques, and the critical events of the corrosion-induced cracking process (first observation of the surface crack) or sudden increases in the crack width. These critical events can only be the result of the development of tiny micro-cracks in the cover zone of the steel reinforcements (due to the expansive character of the corrosion products of steel), and its eventual coalescence producing an open crack reaching the surface of the mortar specimen. In these circumstances, the ultrasonic waves travel through a highly defective medium, leading to a meaningful transference of energy from the fundamental frequency component of the waveform to higher-order harmonics or to the intermodulation products [69].

### 3.2. Possible Influence of Active Mass Transport Processes on the Evolution of the NLU Parameters during the Accelerated Corrosion Tests

It is easily appreciable from Figures 5–8 that, typically, after a clear increase in the nonlinear parameters (or decrease in the case of *DIFA*), the values return to values characteristic of the initial undamaged state (see Figures 5b,c, 6c, 7b,c and 8d,e). This observation was previously pointed out [46–48]. An explanation was hypothetically proposed considering the possible effect of the filling of the void space by liquid containing steel corrosion products after the formation of new cracks or the enlargement of its width [47]. This filling process might be enhanced particularly by the net convective transport of liquid (wick action) in the experimental conditions of this work [47,70]. A recent publication presenting the results obtained during accelerated corrosion tests performed with an experimental setup similar to that used in this work, and using a different NLU technique based on the determination of the sideband peak count-index [71], also concluded that the returning of the nonlinear parameters to values typical of the undamaged state, after the critical events of the micro-cracking phenomena, may be due to the filling of the void space created with liquid containing steel corrosion products. Another possible effect that might contribute to these shifts in the nonlinear parameter values is the fact that the opening of micro-cracks (when reaching the surface) may reduce the overall nonlinearity [71]. However, a careful observation of the results presented in the precedent subsection allow us to appreciate that the abovementioned shifting of the nonlinear parameters is observed not only after the appearance (opening) of the first surface crack, but also after sudden increases in the width of a previously existing crack (see Figures 5c, 6c, 7b,c and 8d,e).

The objective of this subsection was to provide experimental evidence of the existence of active net mass transport processes in the course of the accelerated corrosion tests performed in this research.

One of the noteworthy observations obtained during the accelerated corrosion tests of the RCM specimens was the leaching out of a reddish liquid through the surface open cracks, Figure 9. The measured pH value of this liquid was about 2 to 3, in agreement with previous findings [46,49]. The only possible origin of the strong acidity of the leachate was the hydrolysis reaction of the main product of the anodic electrochemical reaction, i.e., the hydrolysis caused by the $Fe^{2+}$ ions [72]:

$$Fe^{2+} \text{ (ac)} + 2\,H_2O \text{ (l)} => Fe(OH)_2 \text{ (s)} + 2\,H^+ \text{ (ac)} \tag{5}$$

The images in Figure 10 clearly demonstrate the existence of a net mass transport of liquid from the bottom of the RCM specimen in contact with water to the upper surface (see Section 2.2). This net transport, which efficiently drags the steel corrosion products away from the rebar, is probably due to wick action [70]. Additional evidence of the mass transport process is the video produced by collecting photos taken by a webcam system focusing on the upper surface of one of the RCM specimens during some part of the accelerated corrosion test: the images were taken during a period of 39 h after the appearance of the first corrosion-induced surface crack, at a rate of one image every ten minutes (see Supplementary Materials [Video S1]).

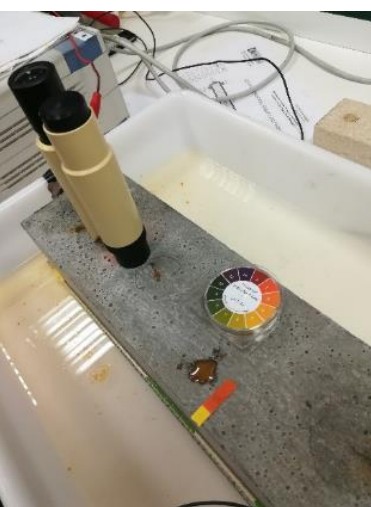

**Figure 9.** Photo showing the liquid leached out through the surface open crack in the course of the accelerated corrosion test. pH value of the liquid: 2 to 3.

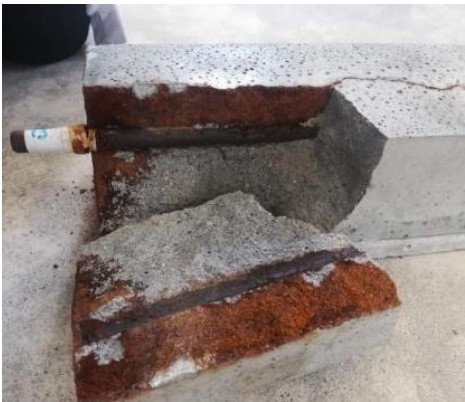

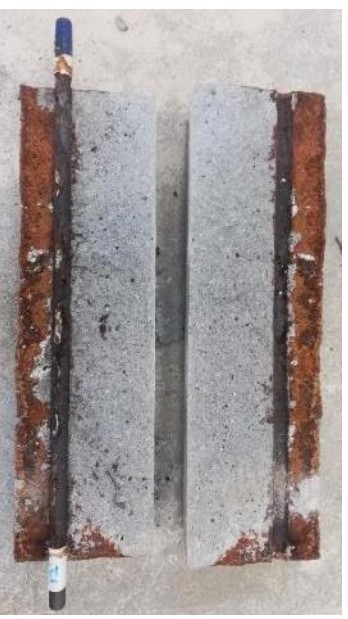

**Figure 10.** Photos showing the appearance of the split parts of an RCM specimen broken after finishing the accelerated corrosion test. The surfaces of the open crack reaching the upper surface are fully coated with the typical reddish-brown corrosion products of steel, while the region below the steel bar is free from corrosion products.

The existence of this active mass transport process during the corrosion tests implies that, most probably, just after the opening of the micro-crack to the upper surface of the RCM specimens, the void space starts to quickly be filled with liquid dragging the soluble and solid corrosion products of steel, finally producing the leaching out of liquid, as shown in Figure 9. A strong dependence of the nonlinear parameters (measurements performed using the Scaling Subtracting Method) on the liquid saturation level of reinforced concrete samples during accelerated steel corrosion tests has been previously recognized [45]. Namely, a significant reduction in the values of the nonlinear indicator when the saturation state was higher, especially for the more corroded states when the cracks reached the surface, was observed [45]. Furthermore, a decrease in the nonlinear elastic response when increasing the saturation level for undamaged concrete samples was also reported [52].

A study using the technique of Nonlinear Impact Resonance Acoustic Spectroscopy (NIRAS) has shown that, in the course of expansion tests of cement mortar bars containing aggregates of different alkali reactivity, the normalized nonlinearity parameter also experiments a complex evolution [39]. For the most reactive aggregates, the nonlinear parameter shows a clear increase during the early phase of cracks incubation (also characterized by the microscopic observation of the appearance of alkali-silica gel), but, later, a post-peak decrease in the nonlinearity coincidental with the phase of crack-width increase was observed. The explanations of the authors for these observations were based on a possible modification of the physico-chemical properties of the alkali-silica gel filling the pores and micro-cracks generated by the expansion process, or to the transport or leaching out of the gel to the surrounding aqueous solution [39]. These observations have some similarities with the shiftings observed in this work for the nonlinear parameters in the course of the corrosion tests, i.e., the increase then decrease in behavior (or decrease then increase in the case of *DIFA*) appreciable in Figures 5–8. However, there is a difference between the findings in this work and those in [39]. The shiftings (increase–decrease) were more abrupt in the case of the corrosion tests (this work) than in the case of the alkali-silica reaction experiments reported in [39]. Figures 5–8 show that the post-peak decrease (or increase) in the nonlinear parameter is observed in only one day, typically at the next measurement after reaching the peak value; while, in the case of the experiments performed in [39], the post-peak decrease took typically 4 to 8 days. A possible explanation for this difference might come from the consideration that the fluid filling the cracks in the case of the steel corrosion test (aqueous solution dragging some iron oxide particles) has very different properties, for instance, a much lower viscosity, when compared to the alkali-silica gel. It is likely that a higher viscosity would slow down any transport process in the case of the gel.

All things considered, it is clear that, in the experimental conditions of this research, there was an active net transport process of moisture from the bottom of the RCM specimens to the upper surface in the course of the accelerated corrosion tests, due to wick action. When a new corrosion-induced crack opens to the surface, it is most probable that the aqueous solution dragging steel corrosion products starts filling the crack, finally producing the leaching out of liquid at the surface. It is reasonable to admit that this process may modify the nonlinear elastic response of the cracked medium, resulting in a shifting of the values of the nonlinearity parameters, which may adopt the values, even characteristics, of an undamaged state. However, the possibility of contributions of other factors to the shiftings of the nonlinearity parameters cannot be ruled out. For instance, it is possible that the fact of the opening of an initially almost closed micro-crack might reduce the nonlinearity [39,71].

### 3.3. Consequences for the Possible Application of the NLU Techniques to Engineering Practices in the Field of the Detection of the Corrosion-Induced Damage of Reinforced Concrete Structures

In dealing with the indirect non-destructive techniques intended to be applied in engineering practices, it is very important to assess the capability of the techniques for an unambiguous identification of the damaged states, and the dispersion of results, which is related to the possibility of reliable quantification through a damage index.

It was previously shown that, in this type of experiment, the higher the damage level, the higher the range of values of the measured values of *R*, but the variability of the results is also higher [48]. The possible causes of this variability were previously discussed [48]. Figure 11 shows the correlations of the *R* values with the crack-width measurements (representing, here, the evolution of the corrosion-induced damage) of specimens M3 and M4, the two specimens that showed an open surface crack during the tests. The high dispersion of the results once the crack reached the surface is appreciable. For this reason, in the previous publication, a statistical treatment was used, first to obtain the UCL of the undamaged state (see Section 3.1) and, later, to obtain statistically significant values of *R* for the stabilized states after finishing the corrosion tests, which were able to provide a clear distinction between the different damage states accrued by the RCM specimens due to steel corrosion [48]. Another possibility for quantifying the damage is the use of cumulative plots of the nonlinear parameters [39,71]. Figure 12 depicts the evolution of the accumulated *R* values (calculated as the time integral of *R*) in the course of the tests of the M0, M1, M2, M3, and M4 specimens. The reference M0 specimen (not subject to passing current) only showed a moderate increase in the cumulative *R* parameter. This behavior is clearly distinguishable from those of specimens M1 and M2, which were subject to the accelerated corrosion test during 3 and 6 days, respectively, but neither of them arrived to show a surface crack. Finally, the M3 and M4 specimens, which cracked at the surface on the 6th day, and were subject to passing current during 6 and 20 days, respectively, showed a marked increase in the cumulative *R* parameter. These observations point out the possibility of using such a type of cumulative plot of the nonlinear parameters for a distinctive detection and quantification of the different damage states.

Some consequences can be derived from the abovementioned considerations for the intended use of the NLU techniques in engineering practices, i.e., surveys aimed at detecting the risk or evaluating the damage induced by corrosion in reinforced concrete structures. Surface micro-cracking seems to be preceded by the significant shiftings of the values of nonlinear parameters, as compared to the undamaged state, as shown here and in previous publications [42–48,71]. However, the high dispersion of the values obtained when NLU measurements are obtained in specimens with a certain degree of damage (visible surface cracks), hinders the possibility of using short surveys based on the attainment of a few values of the nonlinear parameters. Most probably, it would be necessary to perform detailed, longer surveys with the acquisition of a statistically significant amount of data, or use cumulative plots of the nonlinear parameter values. In addition to this, the need to obtain representative values of the undamaged state, either initially (before starting the damaging process) or by testing the structure at non-damaged locations, must be considered. Other particularities related to the complexity of the reinforced concrete structures, for instance, the presence of coarse aggregates and zones with the accumulation of several steel bars may hinder the interpretation of the results of the NLU measurements. All these considerations point out the convenience of using embedded monitoring systems allowing the remote detection of the risk of corrosion and a tracking of the evolution of the micro-cracking process. Another point to be considered is the interest in also using non-destructive techniques, such as the electrical resistivity mapping of the reinforced concrete structure [73], in order to detect the variations in the moisture state of concrete and the possible existence of active moisture-transport processes, which seem to contribute to the variability of the nonlinear elastic response of reinforced concrete specimens subject to corrosion.

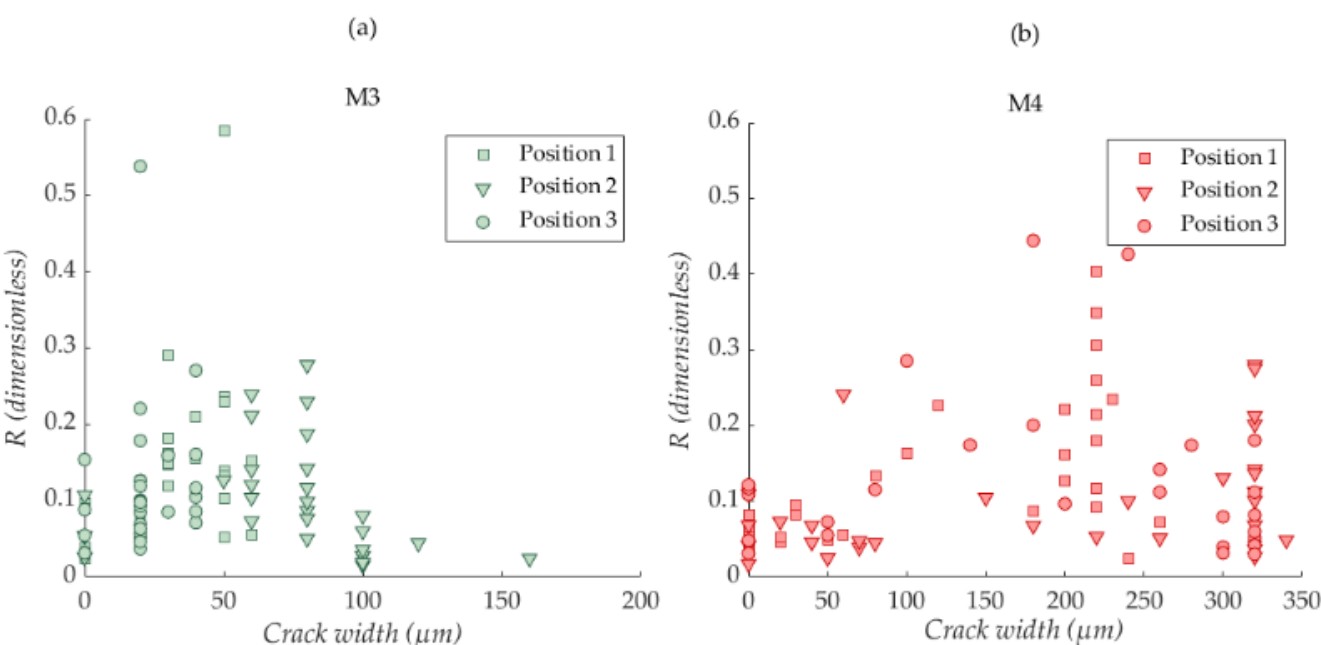

**Figure 11.** Correlations between the values of the *R* parameter and the crack-width measurements obtained during the accelerated corrosion tests. (**a**) M3 specimen and (**b**) M4 specimen. The results correspond to the three positions of testing on the upper surface of the specimens (see Figure 4).

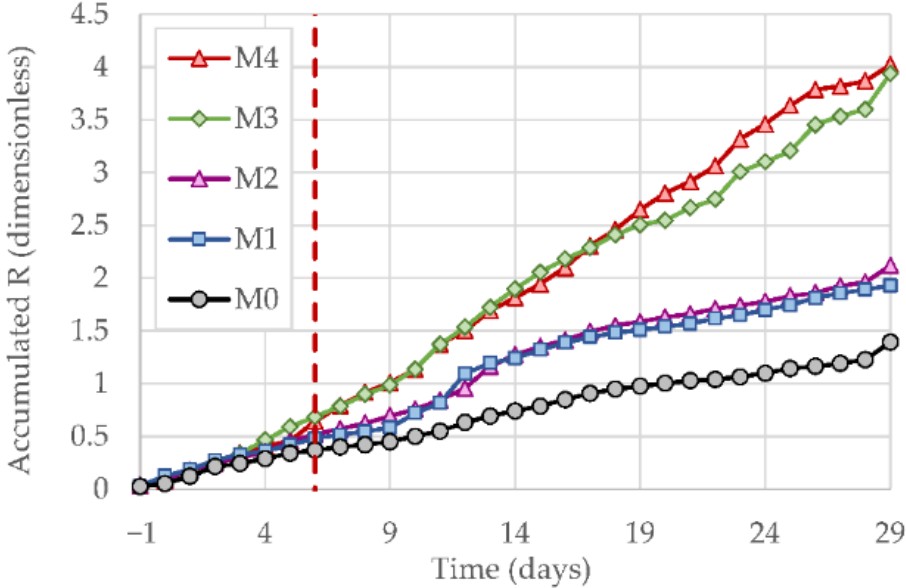

**Figure 12.** Accumulated values of the *R* parameter for the accelerated corrosion tests of the M0, M1, M2, M3, and M4 RCM specimens. The vertical line indicates the time of surface cracking for the M3 and M4 specimens; the rest of the specimens did not show any surface cracks.

*3.4. Influence of the Electrochemical Parameters Governing the Activity of the Steel Reinforcement Corrosion on the Suitability of the NLU Techniques to Assess the Corrosion-Induced Damage*

The elastic wave methods, in this case NLU techniques, have been shown to be very useful for assessing mechanical properties and for detecting damage in materials and constructions. In the current research, we applied these latter techniques to obtain information on the micro-cracking process of reinforced cement mortar produced during the propagation period of the steel reinforcement corrosion process (Tuutti's service life model of corroding structures [74]). The corrosion rate ($I_{corr}$) is a key parameter for assessing the activity of the corrosion process during the propagation period, and this variable

is determined through electrochemical techniques, for instance, the linear polarization resistance (LPR) method [10], electrochemical impedance spectroscopy (EIS) [75], or the methodologies based on applying a galvanostatic pulse (GP) [76]. Our NLU study did not focus on determining the corrosion activity; rather, it focused on studying the mortar's cracking during accelerated corrosion tests performed under a constant (high) corrosion rate in order to obtain information with reasonably short testing periods. It would be convenient to obtain extensive data, with tests performed at different corrosion-rate values, to correlate the NLU results with the $I_{corr}$ and other electrochemical parameters. In particular, it is necessary to demonstrate the suitability of the proposed techniques in natural (non-accelerated) corrosion tests, for instance, corrosion experiments of reinforced concrete specimens exposed to a humid and salt-laden environment, without the application of an electric field. These latter conditions are characterized by a lower value of $I_{corr}$ than those applied in this work [4,5]. However, this was out of the scope of our present research project. The search of such correlations would be an interesting subject for possible future researches.

## 4. Conclusions

The results presented in this work allow us to unequivocally correlate the observations of the strongly enhanced nonlinear elastic features of the received ultrasonic wave with the critical events of the corrosion-induced cracking of reinforced cement-mortar specimens. The nonlinearity parameters, i.e., the third harmonic ratio ($\beta'$) in harmonic distortion experiments, and the intensity modulation ratio ($R$) in intermodulation distortion tests, clearly exceed the threshold values corresponding to the undamaged state some days before the observation of the appearance of the first surface crack, or before sudden, strong increases in the crack width. In the case of the *DIFA* parameter, it reached values close to zero before the relevant cracking events. These observations, which are easily appreciable in plots combining the evolution of the nonlinear parameters and the evolution of the derivative of the crack width with time, point out the possibility of the early detection of cracking using the NLU techniques.

Experimental evidence of the existence of active net mass transport processes, due to wick action, in the course of the accelerated corrosion tests (in the experimental conditions of this work), was presented. Moisture transported from the bottom of the specimen moves through the pore network of the mortar and cracks to the upper surface. This convective transport efficiently drags soluble and solid corrosion products of steel through the corrosion-induced open cracks to the upper surface. This phenomenon is most probably one of the causes of the observed abrupt shifting of the values of the nonlinear parameter values, i.e., increase and then post-peak decrease in $\beta'$ and $R$, (or decrease and then increase in the case of the *DIFA* parameter). The post-peak values show a high variability and can even adopt values typical of the undamaged state.

From the abovementioned observations, some consequences can be derived from the point of view of the use of the NLU techniques in engineering practices, i.e., in surveys aimed at evaluating reinforced concrete structures affected by corrosion. Most probably, it would be necessary to perform detailed, lengthy surveys with the acquisition of a statistically significant amount of data, or using cumulative plots of the nonlinear parameter values. In addition to this, the need to obtain representative values of the undamaged state, either initially (before starting the damaging process) or by testing the structure at non-damaged locations, must be considered. All these considerations point to the convenience of using embedded monitoring systems allowing for the remote detection of the risk of corrosion and tracking the evolution of the micro-cracking process.

**Supplementary Materials:** The following are available online at https://www.mdpi.com/article/10.3390/cmd3020014/s1, Video S1: Leaching out of liquid during accelerated corrosion test.

**Author Contributions:** Conceptualization, M.-Á.C. and J.R.; Data curation, M.M., J.-N.E. and P.P.; Formal analysis, M.M., J.-N.E. and P.P.; Funding acquisition, M.-Á.C. and J.R.; Investigation, M.M., J.-N.E., P.P., G.d.V. and E.-G.S.; Methodology, M.-Á.C., J.-N.E., P.P. and J.R.; Project administration, M.-Á.C. and J.R.; Resources, M.-Á.C., J.-N.E. and J.R.; Software, P.P., G.d.V. and E.-G.S.; Supervision, M.-Á.C., J.-N.E. and J.R.; Validation, M.-Á.C., J.-N.E. and J.R.; Visualization, M.M. and P.P.; Writing—original draft, M.-Á.C.; Writing—review and editing, M.-Á.C., J.-N.E., P.P. and G.d.V. All authors have read and agreed to the published version of the manuscript.

**Funding:** This research was funded by the Spanish Agencia Estatal de Investigación (grant code BIA2016-80982-R) and by the European Regional Development Fund (grant code BIA2016-80982-R). M.M. acknowledges a pre-doctoral fellowship from the Spanish Ministerio de Educación, Cultura y Deporte (FPU16/04078).

**Institutional Review Board Statement:** Not applicable.

**Informed Consent Statement:** Not applicable.

**Data Availability Statement:** The data presented in this study are available on reasonable request from the corresponding author.

**Acknowledgments:** We would like to thank Carmen Andrade for advice on the details of the corrosion testing. We thank also Lafarge-Holcim Spain for providing the cement samples for preparing the reinforced mortar specimens.

**Conflicts of Interest:** The authors declare no conflict of interest. The funders had no role in the design of the study; in the collection, analyses, or interpretation of data; in the writing of the manuscript, or in the decision to publish the results.

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
