# Peer review of "Early Detection of Corrosion-Induced Concrete Micro-cracking by Using Nonlinear Ultrasonic Techniques: Possible Influence of Mass Transport Processes"

_cmd, doi:10.3390/cmd3020014_

Round 1
Reviewer 1 Report
The authors present observations and correlations between the nonlinear features of ultrasonic waves. The following comments are recommended to be considered:
- This research paper lacks in the sense that all of the experimental data have already been previously presented. So actually Section 2 is only a summary of previously published research. Hence the authors need to be better clarify the novelty and significance of this manuscript.
- A methodology flowchart could be added to show the process followed in this research,
- Equations 1&2: Justify why it is an accurate assumption to only consider first-order and second-order intermodulation products
- Equation 4: It is suggested to better describe why this equation has been used.
- Figure 5: horizontal lines: how was this arbitrary limit defined?
- Section 3.2: Could you suggest a proposal (equation) to evaluate this the effect of active mass transport processes on the evolution of the NLU parameters?
- It would be good if in addition to the test results of [47,48], your conclusions were further quantitively supported by additional literature data.
Author Response
REVIEWER#1
Comments and Suggestions for Authors
The authors present observations and correlations between the nonlinear features of ultrasonic waves. The following comments are recommended to be considered:
Thank you to Reviewer#1 for his/her comments and observations.
- This research paper lacks in the sense that all of the experimental data have already been previously presented. So actually Section 2 is only a summary of previously published research. Hence the authors need to be better clarify the novelty and significance of this manuscript.
Answer: The Reviewer is right in the sense that the main raw data (values of the nonlinear parameters and their evolution with time) were presented previously in refs. [47, 48]. However, we honestly think that the present manuscript enjoys a considerable level of novelty due to the following reasons:
(a) We have presented the data in a new form, which, as far as we know, was never used before. We have calculated the derivative of the crack width with time (dw/dt) (Equation 4). Then we have prepared graphs which combine the evolutions with time of dw/dt and those of the nonlinear parameters, (Figs. 5, 6, 7, 8). This helps very much in detecting the correlations between the observation of nonlinear features with the critical events of the cracking process revealed by significant increases of dw/dt. Please, note that these correlations require a continuous monitoring (practically daily measurements) of the crack width. This is also a distinctive characteristic of our work in comparison with other related researches.
(b) We have provided in Section 3.2 new (non-published before) evidence of the existence of active net mass transport processes, see Figs. 9 and 10 and the video included as the supplementary material S1. A detailed discussion is provided in Section 3.2 about the possibility that this transport process might influence the evolution of the values of the nonlinear parameters, including comparisons with results of other researchers [39, 45, 52, 71].
(c) Section 3.3 provides a new (non-published before) discussion of the possible problems that may appear when shifting the nonlinear ultrasonic techniques to real-practice engineering surveys of reinforced concrete structures affected by steel corrosion. The discussion is based on the information so far gathered about the characteristic evolutions and dispersion of the values of the nonlinear parameters.
- A methodology flowchart could be added to show the process followed in this research,
Answer: We honestly think that Section 2 (Materials and Methods), with its four subsections, provide detailed and extensive information on the materials and samples tested, the experimental procedures and the techniques used, especially the nonlinear ultrasonic (NLU) techniques applied to the detection of the corrosion-induced damage.
In our opinion the description of the methodology provides sufficient detail to allow others to replicate and build on the published results, as requested by the Instructions for Authors of the journal. Please, note that two series of tests were performed with different settings of the experimental details (for instance different duration of the passing current period), and different NLU techniques. Hence, it would be practically impossible to prepare a single flowchart showing all the methodological details. The preparation of several flowcharts seems unpractical to us, and would suppose increasing unnecessarily the number of figures and the length of the manuscript.
- Equations 1&2: Justify why it is an accurate assumption to only consider first-order and second-order intermodulation products
Answer: The appearance of nonlinearity symptoms (generation of higher harmonics or intermodulation products) depends on the physical nature of the medium through which the waves are propagating. In our case, the mechanical characteristics of the cracks determine largely the nonlinearity symptoms. It might be possible that higher intermodulation products (order > 2) were also generated. However, a limitation in the detection appears due to two reasons: (i) the nature of the experimental setup used for the NLU measurements (mainly the frequency range of sensibility of the transducers); (ii) the high attenuation of materials like cement mortar or concrete, which is known to be higher the higher the frequency of a signal (this means that it would be very difficult to guarantee the accurate measurement of the amplitude of the higher order intermodulation products). For these reasons we decided to use only the first and second order intermodulation products, i.e. to consider only the frequencies for which the amplitude measurement is ensured to be reliable. The nonlinearity parameters defined in such a way proved to be sensitive enough for the purposes of this work as shown in Section 3.1.
- Equation 4: It is suggested to better describe why this equation has been used.
Answer: Equation (4) defines how we have calculated the derivative of the crack width with time (dw/dt). These data have been used to produce the combined graphs presenting together the evolutions with time of dw/dt and those of the nonlinear parameters (Figs. 5, 6, 7, 8). It was decided to use this type of graphs to help ascertaining if the variations and shifting of the nonlinear parameters can be correlated with the critical events of the corrosion-induced cracking or not. The graphs clearly showed that the values of the non-linearity parameters, i.e. the third harmonic ration (β’) in harmonic distortion experiments, and the intensity modulation ratio (R) in intermodulation distortion tests, exceeded the threshold values corresponding to the undamaged state some days before the observation of the appearance of the first surface crack, or before sudden strong increases of the crack width.
- Figure 5: horizontal lines: how was this arbitrary limit defined?
Answer: The dotted horizontal line in Fig. 5 represents the threshold considered as indicative of a critical increase of the non-linear elastic features: i.e. a ten-fold increase of the initial value of β’ recorded at the undamaged state. This threshold was empirically determined during our first exploratory experiments of this research [46], and later confirmed by more extensive tests [47].
- Section 3.2: Could you suggest a proposal (equation) to evaluate this the effect of active mass transport processes on the evolution of the NLU parameters?
Answer: Section 3.2 discusses the possible effect of active mass transport processes (moisture transport due to wick action) on the evolution of the NLU parameters during the accelerated corrosion tests in the experimental conditions of this work. The existence of this active mass transport process implies that most probably just after the opening of the micro-crack to the upper surface of the reinforced cement mortar specimens the void space starts quickly to be filled with liquid dragging the soluble and solid corrosion products of steel, and this can qualitatively explain the sense of the variations observed for the NLU parameters. This explanation is also compatible with the observations presented in other related researches [39, 45, 52, 71]. However, at the moment it is not possible to provide an analytical description (equation) able to quantitatively describe the abovementioned possible effect.
- It would be good if in addition to the test results of [47,48], your conclusions were further quantitively supported by additional literature data.
Answer: Regarding the correlation between the NLU observations and the corrosion-induced cracking process (Section 3.1), very similar observations, interpretation and conclusions have been described in a very recent publication presenting results obtained during accelerated corrosion tests performed with an experimental setup similar to that used in this work, and using a different NLU technique based on the determination of the sideband peak count-index [71].
Regarding the possible effect of active mass transport processes on the evolution of the NLU parameters, the interpretation given in Section 3.2 is compatible with the observations presented in other related researches [39, 45, 52, 71].
Reviewer 2 Report
The submitted manuscript concerns a research on a non-destructive technique for corrosion development assessment in reinforced concrete structures.
The authors are commended for the excellent work. It is one of the top manuscripts we have reviewed in the past years. The subject is of interest, the structure is appropriate, the writing is error-free and clear, the reference list is quite complete and the discussion is relevant.
Some minor comments:
L34 Spalling is sometimes generalized, not seldom, in marine structures
In Figure 3 it is not clear if the UPV measurements were carried out between reinforcement and top surface or between reinforcement and bottom surface.
Caption of Figures 5, 6, 7 and 8 shall follow the style of Figure 11: (a)…, (b)…, ….
Further, it is not common information or descriptions of the figures in their own caption. This shall be provided in the text where figures are introduced, or as text after the respective caption if the figures were already introduced (which does not happen as a rule in this manuscript, and it is suggested to be revised as well).
There may be an advice regarding the shifting of this technique to “real practice” conditions. The information coming from readings when several bars and coarse aggregates are present can be not much revealing.
A suggestion:
In future works, authors may consider the making of potentiostatic accelerated corrosion development and compare the evolution in time of NLU with current intensity.
Author Response
REVIEWER#2
Comments and Suggestions for Authors
The submitted manuscript concerns a research on a non-destructive technique for corrosion development assessment in reinforced concrete structures.
The authors are commended for the excellent work. It is one of the top manuscripts we have reviewed in the past years. The subject is of interest, the structure is appropriate, the writing is error-free and clear, the reference list is quite complete and the discussion is relevant.
Thank you to Reviewer#2 for his/her kind comments and observations.
Some minor comments:
- L34 Spalling is sometimes generalized, not seldom, in marine structures
Answer: We agree with the Reviewer. The text in line 34 has been changed:
“and eventual localized or generalized spalling of the concrete cover”.
- In Figure 3 it is not clear if the UPV measurements were carried out between reinforcement and top surface or between reinforcement and bottom surface.
Answer: We have clarified the positions of the transducers:
(a) In the main text (lines 265-267 of the revised manuscript), we have included the sentence: “The positions of the transducers were equidistant to the rebar axis and to the specimen’s lateral surface (same height from the specimen’s bottom), see Fig. 3”.
(b) In the caption of Fig. 3, we have changed the text: “Figure 3. Positions of the transducers in the first series of experiments (top view of the RCM specimen). E1: emitter, R1: receiver. Figure adapted from [47]”.
- Caption of Figures 5, 6, 7 and 8 shall follow the style of Figure 11: (a)…, (b)…, ….
Answer: We have modified the captions of Figs. 5, 6, 7 and 8 following the recommendation of the Reviewer. Please, see the revised manuscript.
- Further, it is not common information or descriptions of the figures in their own caption. This shall be provided in the text where figures are introduced, or as text after the respective caption if the figures were already introduced (which does not happen as a rule in this manuscript, and it is suggested to be revised as well).
Answer: We agree with the Reviewer in the sense that common information does not need to be included in the captions of the figures. However, we think that the figures should be self-explanatory. This is why we have included in each figure the meaning of the vertical and horizontal lines, see for instance Figs. 5, 6, 7, 8 and 12.
Following the recommendation of the Reviewer we have simplified and shortened the caption of Fig. 8, since the information on the transducer used for obtaining these data is fully explained in the main text (lines 465-472 of the revised manuscript). The caption of Fig. 8 now reads:
“Figure 8. Correlation between the evolutions with time of dw/dt (* symbols) and those of the intensity modulation ratio R (circle symbols). (a): Specimen M0; (b): specimen M1; (c) specimen M2; (d): specimen M3; (e): specimen M4. The vertical lines indicate the time of the end of passing current, and the time of observation of the first surface crack, which coincide in the case of specimen M3. The dotted horizontal lines correspond to a statistically established upper control limit for the initial (undamaged) R values [48]”.
- There may be an advice regarding the shifting of this technique to “real practice” conditions. The information coming from readings when several bars and coarse aggregates are present can be not much revealing.
Answer: We agree with the Reviewer’s comment. A new sentence has been included in Section 3.3 (lines 663-666 of the revised manuscript). The sentence reads: “Other particularities related with the complexity of reinforced concrete structures, for instance the presence of coarse aggregates and zones with accumulation of several steel bars may hinder the interpretation of the results of NLU measurements”.
A suggestion:
In future works, authors may consider the making of potentiostatic accelerated corrosion development and compare the evolution in time of NLU with current intensity.
Answer: Thank you very much to the Reviewer for this constructive suggestion, which we would try to apply in future work. We also think that the use of a potentiostatic regime for the accelerated corrosion test may provide some advantages related with the information obtained from these tests.
Reviewer 3 Report
The work concerns a very interesting and perspective research tool to assess the destruction of reinforced concrete as a result of corrosion of reinforcement in concrete. However, there are numerous substantive doubts.
- Corrosion of reinforcing steel in concrete is an electrochemical process, therefore, dedicated test methods are known from chemical laboratories, including, among others, polarization methods for the assessment of corrosion damage.
- The use of non-electrochemical methods is possible, but it seems necessary to compare the results of ultrasonic techniques with electrochemical methods. The manuscript does not even mention the possibility of such comparisons.
- Polarization tests, such as the linear polarization resistance (LPR) method or the electrochemical impedance spectroscopy (EIS) method, would make it possible to compare the measured corrosion rates of concrete reinforcement with the results of ultrasonic measurements and perhaps reveal some correlations.
- Assuming that the performed experiments relate to a very advanced stage of corrosion degradation, as a minimum, it seems necessary to compare the ultrasonic measurements with the optical measurements that allow to capture the micro-deformation of the structure of the tested reinforced concrete element caused by corrosion processes.
- To sum up, the conducted experiments revealed no reference to other precise tests performed in parallel, so that the obtained results could not be considered suitable for any verification in the area of the impact of ultrasonic waves.
Author Response
REVIEWER#3
Comments and Suggestions for Authors
The work concerns a very interesting and perspective research tool to assess the destruction of reinforced concrete as a result of corrosion of reinforcement in concrete. However, there are numerous substantive doubts.
Thank you to Reviewer#3 for his/her kind comments and observations.
- Corrosion of reinforcing steel in concrete is an electrochemical process, therefore, dedicated test methods are known from chemical laboratories, including, among others, polarization methods for the assessment of corrosion damage.
Answer: We fully agree with the Reviewer in the sense that the classical electrochemical techniques, such as the half-cell corrosion potential and corrosion current density (Icorr) measurements, are very useful for assessing the risk of corrosion and the intensity of the active corrosion processes. Please, see the text in the Introduction (between lines 40-46 of the revised manuscript). However, we should consider that these techniques are not sensitive to the physical and mechanical damage, i.e. they cannot detect the micro-cracking of concrete due to reinforcement corrosion.
- The use of non-electrochemical methods is possible, but it seems necessary to compare the results of ultrasonic techniques with electrochemical methods. The manuscript does not even mention the possibility of such comparisons.
Answer: Since comments 2 and 3 of Reviewer#3 were closely related we have prepared a combined answer to both comments. Please, see answer to next comment number 3.
- Polarization tests, such as the linear polarization resistance (LPR) method or the electrochemical impedance spectroscopy (EIS) method, would make it possible to compare the measured corrosion rates of concrete reinforcement with the results of ultrasonic measurements and perhaps reveal some correlations.
Combined answer to comments 2 and 3 of Reviewer#3: The elastic wave methods, in this case nonlinear ultrasonic (NLU) methods have shown to be very useful for assessing mechanical properties and for detecting damage in materials and constructions. In our research we have applied these latter techniques for obtaining information on the microcracking process of cement mortar produced during the propagation period of the reinforcement corrosion process (Tuutti’s service life model of corroding structures). The corrosion rate (Icorr) is a key parameter for assessing the activity of the corrosion process during the propagation period, and this variable is determined through electrochemical techniques (for instance polarization resistance). Our NLU study was not focused on determining the corrosion activity, rather it was focused on studying the mortar’s cracking during accelerated corrosion tests performed under a constant (high) corrosion rate in order to obtain information with reasonably short testing periods. It would be very interesting to obtain extensive data, with tests performed at different corrosion rate values, to correlate the NLU results with the Icorr or other electrochemical parameters. However, this was out of the scope of our present research project. The search of such correlations would be an interesting subject of possible future researches.
- Assuming that the performed experiments relate to a very advanced stage of corrosion degradation, as a minimum, it seems necessary to compare the ultrasonic measurements with the optical measurements that allow to capture the micro-deformation of the structure of the tested reinforced concrete element caused by corrosion processes.
Answer: In this work the physical damage of the reinforced cement mortar specimens due to corrosion was followed by detecting the appearance of the first surface micro-crack, and by monitoring the evolution of the crack width (w) over time (daily measurements). The crack width measurements were performed using a microscope. Figures 5 to 8 consist of graphs which combine the evolutions with time of dw/dt and those of the nonlinear parameters. These types of graphs help ascertaining if the variations and shifting of the nonlinear parameters can be correlated with the critical events of the corrosion-induced cracking or not. Hence, we honestly think that the manuscript contains the comparisons between NLU measurements and optical measurements representing the micro-deformation of the mortar specimens.
- To sum up, the conducted experiments revealed no reference to other precise tests performed in parallel, so that the obtained results could not be considered suitable for any verification in the area of the impact of ultrasonic waves.
Answer: In our opinion the results presented in this work allow to unequivocally correlate the observation of strongly enhanced nonlinear elastic features of the received ultrasonic wave with the critical events of the corrosion-induced cracking of reinforced cement mortar specimens, as shown in Figs. 5 to 8. This was one of the main objectives of the work. This research field can be considered now as being only at an exploratory initial level, with many aspects being actually poorly known. The investigation of correlations of the results of NLU measurements with other variables is a field of possible interesting future researches.
Round 2
Reviewer 1 Report
Authors have responded to the comments.
Author Response
Comments and Suggestions for Authors
Authors have responded to the comments.
Answer: Thank you to Reviewer#1 for his/her comments.
Reviewer 2 Report
All comments were properly addressed.
Reviewer 3 Report
The corrosion phenomenon of reinforcing steel in concrete is electrochemical. The use of methods other than electrochemical in experimental research is possible, but must refer to the nature of the phenomenon, i.e. electrochemical parameters. The omission of this aspect in correlation analyzes can, in my opinion, be considered in the category of error. Referring only to the observation of the effects of mechanical damage is, in my opinion, insufficient.
Author Response
Comments and Suggestions for Authors
The corrosion phenomenon of reinforcing steel in concrete is electrochemical. The use of methods other than electrochemical in experimental research is possible, but must refer to the nature of the phenomenon, i.e. electrochemical parameters. The omission of this aspect in correlation analyzes can, in my opinion, be considered in the category of error. Referring only to the observation of the effects of mechanical damage is, in my opinion, insufficient.
Answer: Thank you to Reviewer#3 for his/her comments and observations. We agree with the Reviewer in the sense that more research is needed before ascertaining the suitability of the nonlinear ultrasonic (NLU) techniques for assessing the corrosion-induced damage produced in reinforced concrete structures. Among the aspects to be investigated it is necessary to include the correlations between NLU measurements and the electrochemical parameters governing the activity of the steel rebar corrosion process, especially the corrosion rate (Icorr). However, it should be taken into account the limitation of our research project regarding resources, personnel, time and funding, which precluded extending our work to other aspects such as the abovementioned correlations between NLU measurements and electrochemical parameters. We have added a full new subsection in order to explain the readers the need of further research to investigate these correlations and other aspects which are still scarcely known. Please, see subsection “3.4. Influence of the electrochemical parameters governing the activity of the steel reinforcement corrosion on the suitability of the NLU techniques to assess the corrosion-induced damage” in the revised manuscript.
Round 3
Reviewer 3 Report
I understand the limitations of a research program and that all aspects of a given research problem cannot be analyzed. I welcome the addition of point 3.4 in the manuscript, relating to the electrochemical aspects of corrosion testing of reinforcing steel in concrete. However, there is no indication of specific names of polarization methods that are used in testing the corrosion rate of steel in concrete, ie EIS, LPR and GP. Secondly, there is also no comment on the stages of degradation of reinforced concrete structures. It should be emphasized that the method used in the manuscript makes it possible to identify the development of corrosion of the reinforcement in concrete only in the last, most dangerous stage of degradation. At this stage, the costs of reinforced concrete repair are the highest. Using electrochemical methods (EIS, LPR and GP) it is possible to identify the development of corrosion at a very early stage, when the effects in the form of cracks in the concrete cover are not yet present.
Author Response
Answer: We have introduced a sentence in the Introduction to indicate the capability of the electrochemical techniques to detect the corrosion of steel in concrete before the concrete cover gets physically damaged; we have enlarged a sentence in Section 3.4 mentioning the three most known techniques for determining the corrosion rate in reinforced concrete structures (LPR, EIS and GP); and finally we have introduced two new references [75, 76]. The changes are highlighted in green